# Gudáang 'láa Hl ḵíiyanggang: I Am Finding Joy in Haida Repatriation and Research

## Lucy Bell Sdahl Ḵ'awaas

Haida Nation, Haida Gwaii, V0T 1M0 and Individualized Interdisciplinary Studies, Simon Fraser University, Burnaby, BC V5A 1S6, Canada; haidalucy@gmail.com

**Abstract:** Over 12,000 Haida belongings and 500 Haida ancestral remains were collected and locked away in museums at the height of colonization in the late 1800s to early 1900s. It has been my lifelong quest to undo the colonial harm done to my Ancestors and their belongings. With gudáang 'láa, (joy) as a foundational philosophy and methodology, I am researching and telling the story of Haida repatriation and reconciliatory work with museums.

**Keywords:** Indigenous art history; museology; Northwest Coast; Haida; Indigenous methodology; joy; repatriation; healing; museum practice





## 1. Ḵats'ahlaa—Come in

Welcome to my reflection on finding joy in Haida repatriation and research. Sdahl Ḵ'awaas hinuu dii kya'aang, dii uu X̱aadáagang. As a Haida researcher from the Tsiits G'itanee Eagle Clan and a museum professional, it is important for me to find gudáang 'láa (joy) in all that I do. I have been repatriating, collaborating with, and working in museums for almost thirty years, and I am now conducting my PhD on Haida museology. It has been a privilege to repatriate Ancestral remains, to be surrounded by the thousands of ancestral belongings in museums and to share in the wonder of Haida creativity with my friends and colleagues on the Haida Repatriation Committee. It is in my genes to seek to be k'adangáa (wise) and to blend my Haida ancestral knowledge with my academic knowledge and professional life. It is also in my genes to find and show joy, humour, love, and resilience in my research and work with museums.

The Haida come from Haida Gwaii, a small island on the Northwest Coast, of BC and southeast Alaska. Between 10,000 to 20,000 Haidas once lived in many villages, until smallpox brought the population to approximately 600 and the Haida congregated in the main villages of G̱aw Tlagée (Old Massett), HlG̱aagilda (Skidegate) and Hydaburg, Alaska. It must have been an intense time of grief and survival for the remaining 600 people who were trying to navigate the great changes while holding onto traditions. Generations later, we still honour our Ancestors' ways. We are matrilineal, inheriting our rights and privileges, including crests and traditional names from our mother's side. X̱aad Kil/X̱aayda Kil is a language isolate and the Haida are keenly speaking life back into our once silenced language. Like all Indigenous peoples, the Haida have endured great devastation due to colonial endeavors but are still here, creating, singing, harvesting, laughing, and fighting for what is inherently ours.

At the height of colonization in the 1800s, diseases, the Fur Trade, the Anti-Potlatch law (1885 to 1951), and the heavy church influence contributed to the loss of over 12,000 Haida belongings and 500+ Ancestral remains. Many global museums were influenced by modernist primitivism, a Eurocentric view of "art" created by the "other" and were competing for the best and biggest collections of Northwest Coast that included belongings from charms to housefronts and human remains (Cole 1985). Haida belongings, known for their distinctive and beautiful style were collected, sold, or stolen by collectors, missionaries, Indian Agents, museum collectors and sometimes Haidas. Many collectors were conducting "salvage

anthropology", thinking Indigenous people were heading towards extinction, collecting "artifacts" of what they thought were a "dying people". The scramble for Northwest Coast belongings between the 1800s and early 1900s led to the majority of the Haida collection ending up in museums in Victoria and Vancouver, BC; Hull, QB; New York, NY; Chicago, IL; England and Germany. Most of the Haida collection has been sitting on museum shelves, never to be seen by the general public. In most museums, less than 3% is ever shown in exhibitions, like the belongings I am showing in Figure 1. Haida belongings in museum collections are seen as "art", put out on display by museums without much context or connection to the original owners, yet they are personal and clan property, originally created and shown off in daily activities or Potlatches.

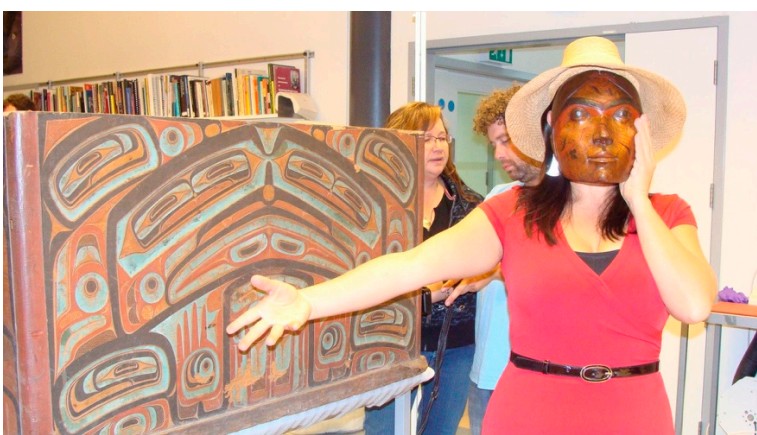

**Figure 1.** Lucy Bell shows bentwood box at the Pitt Rivers Museum. Photograph by Vince Collison.

The loss of the 12,000 Haida belongings that now sit, collecting dust on museum shelves and the remains of 500 kuníisii (Ancestors) we repatriated is the focus of my work and studies. The Haida are on the frontline fighting for Indigenous repatriation and righting colonial wrongs in museums. It has not been an easy journey. We still feel the impacts of colonial harms, and we still face racism and anti-Indigenous museum policies and procedures. I unfortunately endured great harm when I worked at a museum (SFU 2021). Our resilience, strength and hope are powerful antidotes to colonial and racist attitudes and the hoarding of Indigenous belongings and Ancestral remains.

Fogarty et al. say Indigenous peoples are often framed in a narrative of negativity, deficiency, and failure (Fogarty et al. 2018, p. 2). Yáats X̱aadée, ("white" people) studying Indigenous people often have this deficit approach. They often feel the need to provide solutions to "Indian problems", and this attitude is reflected in their research approach, discourse, and daily work. This often leads to racist policies and practices. My museum work and research are the opposite. I am participating in and researching how Indigenous people and museums are addressing the historic "colonial problems" that still impact us today. Do not get me wrong, I do feel the weight of colonial oppression and I am not attempting to downplay the devastations; I just choose to be hopeful and focus on joys in my work.

## 2. Aajii hlan-gwaay.uu ya'aats' gingaan uu giidanggwa—The World Is as Sharp as a Knife

In Haida, there is a philosophy: the world is as sharp as a knife. The way I conduct myself as a Haida museologist and my research methodology are important. My intentions must be good. I must not disrupt the balance, or I will cause harm. I must walk carefully on the edge of the knife and perform things in a Haida way to make a good impact.

By immersing myself in a Haida way of conducting myself and doing research, I have allowed myself to come to a deeper understanding and connection to my kuníisii and our worldview. I continue to abide by the traditional Haida methodology I describe in my X̱aad

kil (Haida language) research (Bell 2016). In the academic world, my approach may be referred to as a blended theoretical approach since I combine a Haida approach (situating myself in traditional values, conducting ceremonies, offering prayers, connecting to healers, and listening to my dreams) with auto-ethnography, archival research, and participatory action (Chilisa 2012).

A Haida methodology is based on traditional values. Some of the traditional values I incorporate in my museology research are yahgudán*g* (respect), ḵangang (responsibility), kil 'laa (kind words), and gudáang 'láa (joy). Basing my methodology on values may seem simple but it is complex and takes time and commitment. Following a values-based methodology that features joy and humour will help me produce something that is more relatable to a Haida and Indigenous audience and it will help forge strong, personable relationships with museum professionals. It helps me write something more meaningful and not focused on academic theory and jargon that my Ancestors and Haida colleagues would not connect with. I will not use terms like "artifact", "skeletons", or "art" that our kun�‘iisii did not use when referring to their property and deceased relatives. These values guide how I conduct research and disseminate my findings.

### 3. Gin gée sdíihlda 'láagang—Return of Property Is Good

One of my biggest joys has been from being a part of the Haida Repatriation Committee (HRC) for three decades. We are repatriation leaders in Canada. We have been negotiating, researching, fundraising, and bringing home the remains of our Ancestors' remains that sat on museum shelves for around 100 years. Standing strongly as the HRC and also connecting to and repatriating our belongings in global museums has been a great honour.

Repatriation has often been traumatizing and overwhelming, yet amazing work to undo the colonial hoarding of our kun�‘iisii's bones and belongings. Situating ourselves in Haida values and protocols has grounded and armed us for the difficult work of repatriation and dealing with colonial and often racist attitudes in the museum world. Remembering the gudáang 'láa (joy) has enabled us to find joy in our challenging museum work. It has been one of our secret weapons to success in repatriation.

The náanlang (grandmothers) have been the backbone of the Haida repatriation success. Náanii (grandmother) Mary Swanson, Ethel Jones, Gertie White and Leona Clow taught us to conduct ourselves with yahgudang (respect) and to remember to cherish the happy moments in our challenging work. It was these women who showed us great joy, respect and love. Candace Weir speaks to their dedication and cheeky outlooks:

> I remember we were sitting at Co-op [grocery store], flogging our merchandise [repatriation committee merchandise, sold to raise funds for repatriation work] one day, and I'm at the table and Náanii Ethel is walking by and she's like "I'm coming! I'm coming!" She's shopping away, and Vince and I are at this table and it wasn't even a hardship or anything. Things like that make the whole process easier with their support…And then at another fund-raiser, Náanii Mary was at the table and it was a big seller and big item one year to get ladies' butterfly thongs, and men's briefs and bikini bottoms and Náanii Mary's really pitching our underwear line! Selling our thong underwear! (Krmpotich 2014, p. 57)

We started traveling to museums as a delegation in 1996 with the náanlang at the helm. There was strength in numbers, and we were an army of repatriators. When we first began bringing home Ancestral remains from museums, I thought we were just going to bring home the bones for burial, and that is it. I thought we were doing this work so the next generation would not have to. Haida repatriation has been more than "just" bringing home Ancestral remains. We were interacting with the thousands of Haida belongings and archival materials in museums from the American Museum of Natural History in New York to the Pitt Rivers Museum in Oxford. We were celebrating the creativity of our Ancestors. We were studying ancient carving, weaving, and regalia-making skills. We were getting to know our relatives and friends from other villages. We were strengthening the liis (the

invisible string) connecting us to the Ancestors and their belongings and incorporating ancestral knowledge into our everyday lives.

I started my work in repatriation before I had my daughter. I naively thought I was doing this work so my daughter and future generations of Haidas would not have to. She was born into repatriation; it is an important part of her life still. Dr. Cara Krmpotich came to live with me and learn about Haida repatriation for her PhD research and received her first lesson on repatriation from my 4-year-old daughter (Guudangée X̲ahl Kil), Amelia Rea. Cara recalls watching cartoons with her on her first morning at our home:

> She explained how they wrapped the bones in cedar mats and why they were bringing the bones home. Lucy is grateful for the love the repatriation naaniis have shown to her daughter and is proud of her daughter's entrepreneurship in service to her ancestors (as young as four, she set up her own sales are in her parents' café selling raffle tickets to raise funds in support of repatriation efforts). (Krmpotich 2014, p. 73)

Amelia went on to be a life-long, dedicated Haida Repatriation Committee member. One of my favorite memories was when the Haida delegation visited the National Museum of the American Indian (NMAI) in 2002. It was our first repatriation trip together. She became known as our "repatriation baby", as shown in Figure 2, and she continues to be drawn to museums and all things Haida. At the NMAI, we were greeted by Indigenous staff and visited the beautiful Haida collection. When we told the staff we wanted to share our songs and dances with them and dance life back into the belongings, they were so open to it. Carts were filled with rattles, daggers, and woven hats for us to choose from. We all chose something to dance with and it was a powerful experience for us. It was an incredible celebration of our Ancestors and their belongings and a great introduction to the museum world for Amelia.

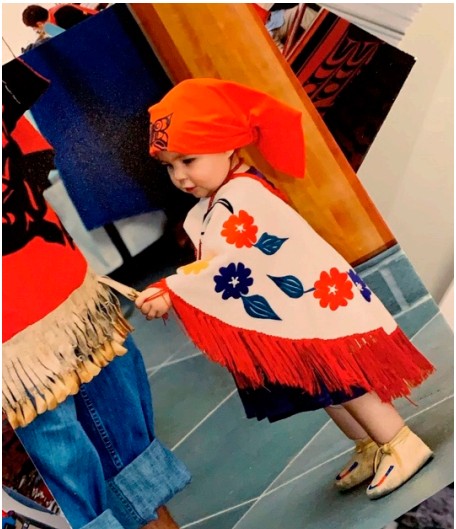

**Figure 2.** Guudangée X̲ahl Kil, repatriation baby at the NMAI. Photograph by Trish Collison.

Twenty years later, I continue to visit Haida collections in museums with my daughter. The two of us were visiting artists\researchers at the Bill Holm Centre at the Burke Museum in 2023. While at the Burke, besides enjoying fried bread from the café every day, we made many discoveries together. In their archives, we found an article about "canoe wings", something we probably have seen in museum collections but have not identified as canoe parts (Emmons 1928). This is important knowledge for modern canoe-makers like Christian White and Jaalen Edenshaw, to whom I sent the article immediately. Amelia and I visited with many Haida belongings at the Burke, including button robes by our relative Hazel Wilson (1941–2016). We loved seeing her use of so many buttons, shells, and coins and sharing our own story about using pennies hammered into buttons by her brother,

S<u>G</u>aaniwáans for a tunic. We also studied the blanket and noticed an old way of securing button blankets that we do not use today as shown in Figure 3. We look forward to trying this old technique on new button blankets.

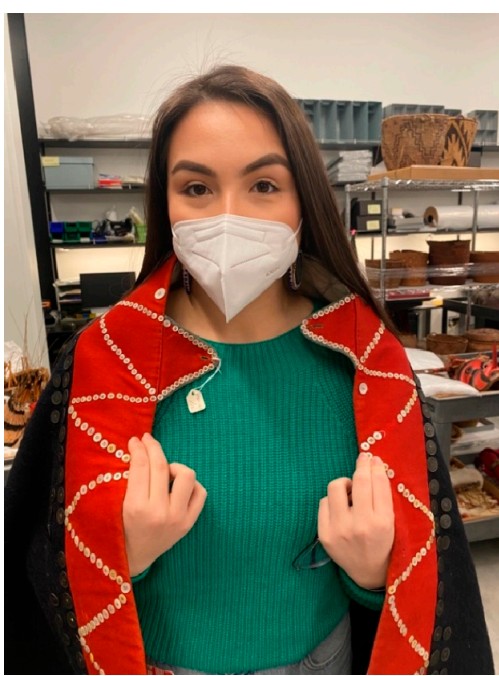

**Figure 3.** Guudangée <u>X</u>ahl Kil trying on a button blanket at the Burke Museum. Photograph by Lucy Bell.

Guudangée <u>X</u>ahl Kil was recently an intern at the UBC Museum of Anthropology (MOA) and hosted a virtual and in-person visit to the collections with 30 Haida participants. The virtual visit was a first for MOA. Connecting the Haida to the collections was important for Guudangée <u>X</u>ahl Kil. She fondly recalls:

> I worked with my community to select thirty belongings from the collection to spend time with, which I then took out of storage and prepared them for the visit. Bentwood boxes, feast bowls, spruce root and cedar-bark baskets and hats, masks, frontlets, tools, clubs, and jewelry. I did my best to select a variety of treasures. The community visits were full of magic, inspiration, and learning. I'm grateful and proud to have brought my people here and I hope this strengthens our relationship with the Museum of Anthropology. (Rea 2023)

Other Haida Repatriation Committee members have grown artistically from our museum visits and shared their good experiences with me. Jaalen Edenshaw and Christian White are two master carvers and keen researchers. Vern Williams Jr. is a master song composer, singer, and musical instrument-maker. Dorothy Grant is a master fashion designer with a love of historic Haida clothing and regalia. They all shared exciting museum discoveries.

Jaalen and Gwaii Edenshaw, the "twindians", are master carvers who have been studying ancient bentwood boxes in museums and archives. Jaalen shared his experience of being commissioned by the American Museum of Natural History and the Pitt Rivers Museum to create reproductions of bentwood boxes in the collections as in Figure 4:

> You can see where other people have been inspired by similar things. The more exposure you have the more options you have when you make decisions when you carve a piece. When you're first starting out, you might only do a split in the Us because that's all you know. You start seeing someone start this here, and then he's put these little crescents in there, which are not 100% unique, but it's just a

different way to do it, to break up that space. And then once you see it, it sort of gives you the option of incorporating into your stuff. (Edenshaw 2021)

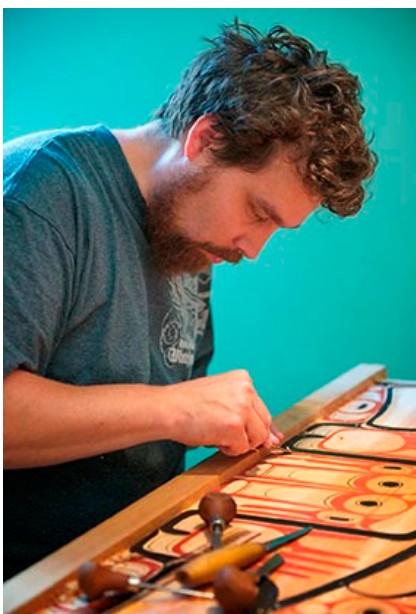

**Figure 4.** Jaalen Edenshaw replicating a bentwood box. Photograph courtesy of the Pitt Rivers Museum.

Master carver Christian White proudly trained his apprentices to make hundreds of bentwood boxes for the Ancestral remains to be buried in as shown in Figure 5. He also traveled to many of the museums to accompany the Ancestors home. He recalls his experience:

> When we see these objects in different museums, we can bring what we learn from that back to our community and have commissioned pieces from artists to bring that back into our ceremonies. It's new for us because the practice of carving has almost been forgotten. And so now we're seeing the value of bringing these back, renewing the songs, the dances, the ceremonies, and so almost everything that we do see in the museums are connected with our culture and our ceremonies. (White 2021)

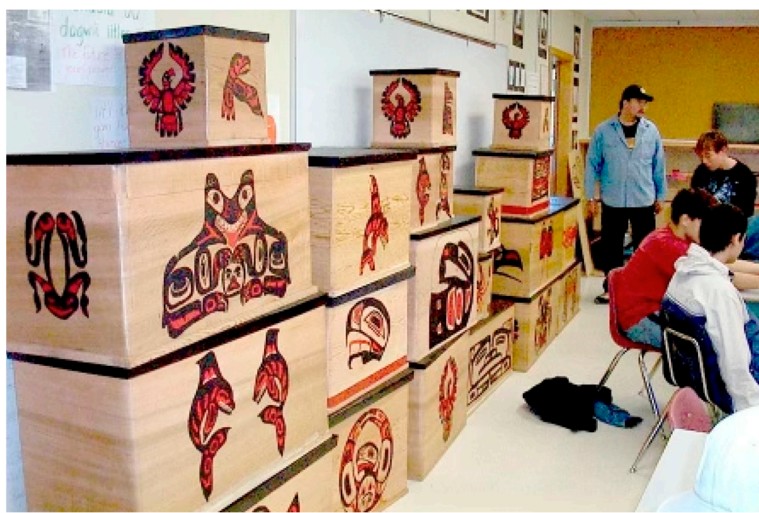

**Figure 5.** Christian White leading youth in traditional bentwood box-making. Photograph by Vince Collison.

Christian also acknowledges the similarities between two Haida carving mediums:

> I was very lucky to actually go in, have an opportunity to look at some pieces that were in collections. Some that I never would have thought about even, how did they make such things as the mountain goat horn spoons, which were very finely carved. They look very similar to the argillite carving. They're shaped so beautifully. (White 2021)

Vern Williams Jr. is a Haida song composer. He also has a love of ancient Haida musical instruments like the flute he made in Figure 6. He helped with the "Listening to the Ancestors—The Art of Native Life Along the North Pacific Coast" exhibit at the National Museum of the American Indian in 2008 by creating traditional rattles for visitors to use. He is also inspired by the songs in archival collections. He studied Haida songs written down by Reverend Harrison (1925). They did not have a recorded tune, and Vern put a new tune to "the berry song". Christian White and his apprentices made new bird masks to accompany the song. The Tluu X̱aada Nee Dance group now shares the revitalized song and dance. Below is the song documented by Harrison:

> Berry Song
>
> Whít squalé squalé, whít squalé squalé, Á lá whít, á lá whít.
>
> Kálungá olthé, kálungá ólthé, Siamzi whé, siamzi whé whít.
>
> Invocation to a bird that ripens berries it is called the Whít, and is beseeched
>
> to produce nicely colored ones, also very large ones
>
> (Harrison 1900, p. 47)

Vern contemplates song revitalization:

> Those songs were created out of a respect and love of wanting to honor what was left for us. And each time we put one of those older songs, the original songs back into our repertoire again, it adds a strength to us and helps us move forward and makes us proud that we're able to bring those treasures back. (Williams 2021)

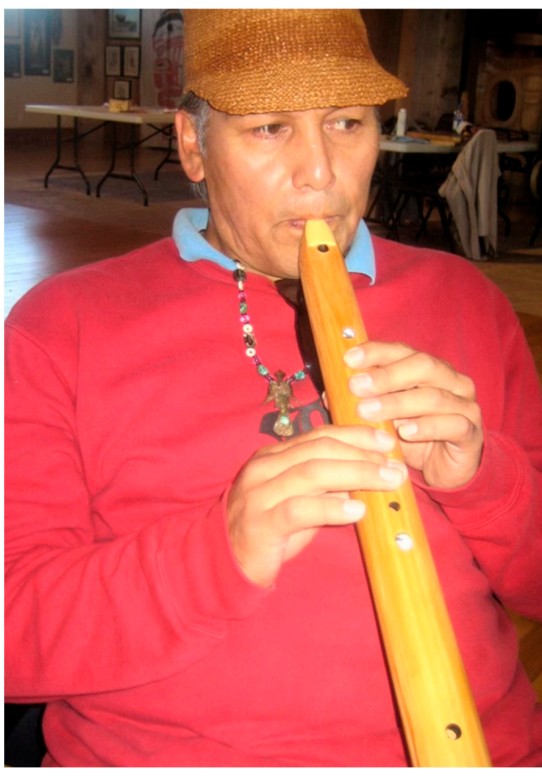

**Figure 6.** Vern Williams Jr. playing a traditional whistle he made. Photograph by Lucy Bell.

In 2023, I went to the National Museum of Natural History (NMNH) in DC. This was my first international museum visit by myself. It was a lonely and strange experience without the support of a bunch of Haidas. I spent the week in the collection and in the archives, looking for interesting stories to share in my dissertation. Without the strength of the delegation, I prayed extra hard and asked my kuníisii for guidance. I asked them to show me joy in the 800+ belongings and archival records. I decided to host a virtual visit for Haidas who could not be by my side on this museum visit. Two dozen Haidas and friends joined me on Zoom. I selected a range of carvings, hats, baskets, gambling sticks, appliquéd clothing, and tools I thought the participants would want to see. Our NMNH colleagues were so supportive of the virtual exchange. We spent half the day looking at the collections, sharing stories, analyzing how things were made and sharing our joy in being with the treasures. A highlight for me was showing renowned Haida fashion designer Dorothy Grant some appliqued belts she was excited to see as she had not seen them before and sharing some spruce-root woven hats, including one by Isabella Edenshaw. Dorothy Grant fondly speaks of Isabella's master weaver talents:

> The things that really, really challenged my brain and artistic ability was the Isabella Edenshaw hats. She doesn't weave over a wooden shape. She free-formed and that takes a lot of balance in your hands and your brain and everything. She was a genius with her hands and her mathematical brain. I came to where I could just tell which ones were hers. Just by looking in a sea of them I could tell her weaving and her husband Charles' signature, a four-star painted in black and red. (Grant 2023)

It was also a highlight to bring out the only known Haida tattoo kit, shown in Figure 7, for everyone to see. We discussed the strong revitalization of crest tattoo designs and stick-and-poke tattooing. That morning I showed only about fifty Haida belongings, many of which had smiling faces painted, appliqued, or carved on them and it was a joy to see the many smiling faces of my friends and family looking on through the computer screen.

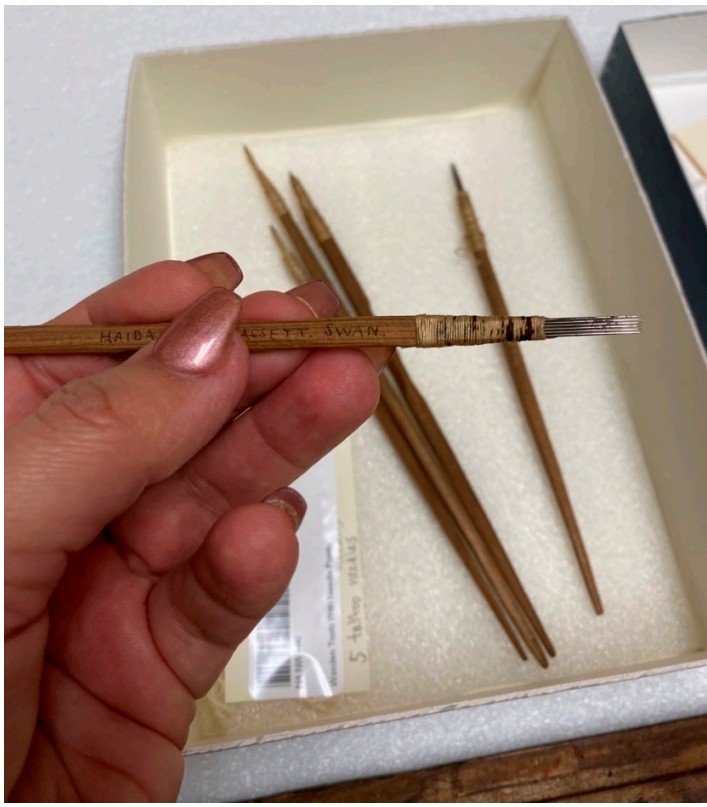

**Figure 7.** Tattoo kit at the National Museum of Natural History. Photograph by Lucy Bell, 2023.

#### 4. Ḵugíin 'láa—Good Paper/Books

In my archival research, I also sought out the writings that would show my Ancestors' resilience, wisdom, and cheekiness. I lost myself in journals of collectors, artists and researchers who visited Haida Gwaii from the 1800s to 1900s. I again would ask my kuníisii to show me their stories and take me on a journey with them. It has been important for me to be intentional with my quest for ancient stories that reflected resilience, humour, and joy. My kuníisii showed themselves and their joy to me through many publications.

My favourite article was written by George Murdock, "Rank and Potlatch Among the Haida, "about my namesake and Ancestor, Sdahl Ḵ'awaas who held a house-building Potlatch at the ancient village of Yaan (Murdock 1936). This is one of the few early publications that portray matrilineal society properly and also featured the wealth of tangible and intangible wealth of my Ancestors. Usually, the historical writings of Haida society were written from a paternalistic perspective by Yáats X̲aadée ethnographers who do not grasp the notion of matrilineal society. In this article, I felt like I was sitting in my Ancestor's 'waahlal (the house-building Potlatch) witnessing my relatives receiving names and tattoos and paying respect to the opposite Raven Clan who helped them birth babies, give tattoos, build their homes, and witness life and rank changes. My favourite part of the story captured humour and joy, something that was not captured by ethnographers. Sdahl Ḵ'awaas gave out many Hudson's Bay blankets in payment to the Raven Clan, who helped her build the home and gave her husband a rag for all of his "hard work". I love how her cheekiness was captured in the story!

James Swan was able to experience and document his own joy and wonder of being amongst the Haida and being on Haida territory. While digging through the NMNH archives, I found two sketches by James Swan from 1883. I had never seen these sketches by him before, so it was a real treasure. In these drawings, Swan showed a mortuary scene featuring the belongings of recently deceased Haidas. Their belongings included carvings, coppers and a robe, with crest designs, a great discovery to make. But what really drew me in was that one person's precious belongings also included a bottle of hot sauce shown below in Figure 8! My ancestors had an appreciation for the technology, food, and gifts from afar, so much so that a deceased person was shown with them. I can think of a few Haidas today who would also want their beloved hot sauce beside them in death.

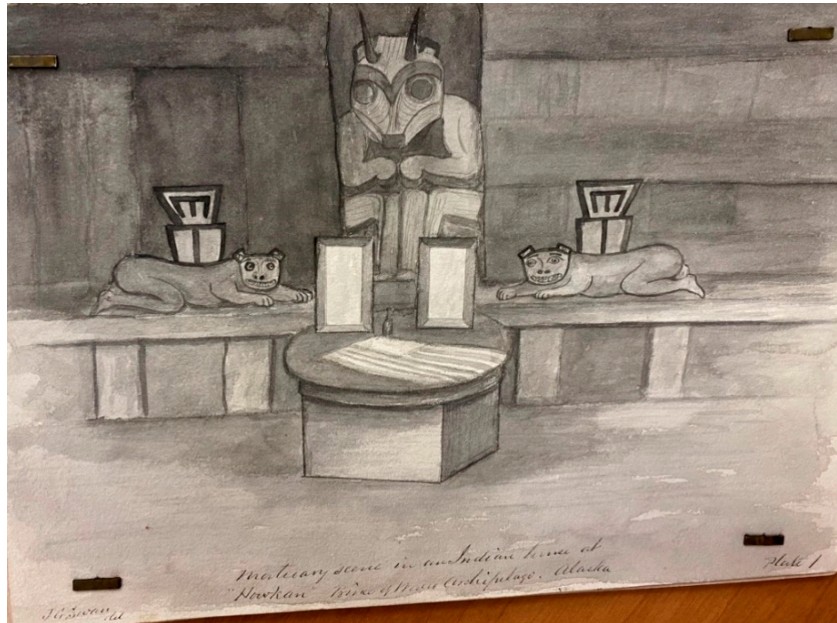

**Figure 8.** James Swan paintings of a Haida mortuary scene with bottle of hot sauce. From the Smithsonian National Anthropological Archives. Photograph by Lucy Bell.

James Swan's Diary of a cruise to the Queen Charlotte Islands was also inspirational, humorous and featured a lot of delicious Haida foods (Swan 1883). Swan's dream of visiting Haida Gwaii was finally realized in the summer of 1883 when he boarded the Hudson Bay Company's Otter. He spent the summer on Haida Gwaii, documenting his excursion in his journal. He spent over a month, hunkering down in a Hudson Bay house with Johnny Kit Elswa and Kit Elswa's cousin, SG̲aana Xaaygens. Swan was in love with traditional Haida foods! His description of the delicious seafood and berries almost makes his journal feel like a manuscript for a food series. He also wrote about the children who brought him berries and geological samples for the Smithsonian collection. This resonated with me because as someone who hosts museum colleagues on Haida Gwaii, I know I would have sent my daughter and her cousins out to collect rocks and berries for them.

In my historical research, I was surprised that Emily Carr's work resonated with me. Some may see her as controversial and may think she was guilty of cultural appropriation. Others, like Haida Repatriation Committee member and carver Gwaai Edenshaw say Carr was "Honest and unique. She was not taking money out of the pockets of the masters whose shadows land in her paints. She was engaged with the community. If anything, her work has increased the reach of our market" (Byfield 2019, p. 1). I could write a whole other essay on the Indigenous views of Carr, but for now I will focus on joy her work has brought me. Besides her haunting paintings of old Haida poles, I was also touched by her journal (Carr 1941). She was a truth-teller, and she captured personal stories of Indigenous people while using her humour and personalizing her experiences on the Northwest Coast. She was in Haida Gwaii in 1912 and 1928 and witnessed the paternalistic attitudes towards women and a great change happening in the world. She too felt the oppression of the church. She was unmarried and was forced to stay with the missionary's family. Carr grumpily said, "The missionary got a farm girl, with no ankles and no sense of humour, to stay there with me" (p. 101). She saw the influence of the church, the emptying of the villages, and the numerous deaths of children. Yet, she was able to find the stories of joy and resilience through all of this. She focused on stories of Indigenous women and children and was able to weave in and out of grief and happiness in a relatable way.

The archives are full of devastating truths and racist attitudes towards Indigenous people. But the archives are also full of joyful anecdotes about our Ancestors if we are open to seeing them.

**5. Ending on a Happy Note**

A healer I saw once had a vision of a náanii sitting on a beach, putting kelp on her head to make people laugh. The healer said the Ancestors want my work to also bring joy. This is important for me to accomplish as a Haida researcher. By focusing on hope, joy and laughter in my research, I not only honour the spirit of my Ancestors but also create a more inclusive and hopeful narrative.

The idea that my kuníisii and my daughter are listening underscores the importance of passing on the wisdom and laughter gained from my research. It is through this intergenerational transmission of knowledge that the lessons learned and stories told can continue to inspire and educate future generations, ensuring the preservation and celebration of Haida culture.

My commitment to joy of Haida research is a powerful reminder that research and academia should not solely focus on academic theories, sad statistics and the tragedies of Indigenous people but also highlight the strength, resilience, and joy that can emerge from those difficult experiences. I hope my research will have a lasting and positive impact on the Haida and that you smiled at least once while reading my story.

**Funding:** This research was funded by Social Science and Human Research Council (SSHRC) #767-2021-1950, and Old Massett Village Council's (OMVC) funding support for my research.

**Acknowledgments:** Haw'aa to the kuníisii for showing me a glimpse of your life and your beautiful belongings. Haw'aa also to the Haida Repatriation Committee for your unwavering dedication to honour our kuníisii and for sharing your joyful stories with me. Haw'aa to Marianne Ignace for your X̱aad Kil guidance. Daláng 'wáadluwaan ahl Hl kíl 'láagang. Thank you all!

**Conflicts of Interest:** The author declares no conflict of interest.

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
