# Peer review of "Gudáang ‘láa Hl ḵíiyanggang: I Am Finding Joy in Haida Repatriation and Research"

_arts, 1940_

Round 1

Reviewer 1 Report

Comments and Suggestions for Authors

This article clearly sets out what it aims to do, highlight the 'joy' in Haida repatriation work. This includes the powerful examples of the author's child in connection to the word 'joy.' The language is accessible, and tangible, and that too, in alignment with Haida methodologies of language. I would suggest that the use of the word 'pioneer' (page 3/14) be assessed in connection to scholarship that challenges its usage in the context of white settler colonial histories.

In addition, there has been some very interesting critique of Emily Carr as well, which the author might want to acknowledge though they may not agree with it. See: https://prenticepieces.com/2019/08/07/when-cultural-appropriation-is-forgiven-the-strange-case-of-emily-carr/ and https://canadianart.ca/microsites/cover-stories/1993-fall.pdf. 

In order to strengthen the robustness of the paper, the author might want to incorporate more sources in the bibliography section which would be useful for other scholars, or students, wishing to learn more about the work of Haida repatriation. 

Author Response

Thanks for taking the time to review my essay. I incorporated your input. I appreciated the article you suggested on cultural appropriation. I also included info from Dr. Cara Krmpotich's research on Haida repatriation.

Reviewer 2 Report

Comments and Suggestions for Authors

Dear author, 

Thanks for the original, respectful, and passionate article about repatriating Haida "belongings" and your participation in such an important activity. 

To start with. Please give a short context (location, size of community, history) of the Haida for those readers that do not know about this indigenous nation. I am sure that in your words this would be a great help to bring awareness about the fact that Haida's were not destroyed, that they are alive and resisting today. That your work is fundamental for the epistemic reconstitution needed after centuries of coloniality, and that such work is part of the process of re-engaging culturally and politically with such realities. 

Please make sure to use a consistent quotation system. Please check to the PDF attached with underlined (yellow) sections and comments about it. 

Thanks for the contribution of Haida research methods, please make note of where required to expand on those, introduce an academic and non-academic discussion on them, and how these methodology complements and/or critiques hegemonic ones (anthropology, ethnography, museology, etc)... Define (in notes) some needed comments, such as "savage anthropology" (which could be of great help for you whole argumentation). 

Best regards,

Comments on the Quality of English Language

Minor revisions, for spelling and consistency in quoting is needed. 

Author Response

Hi,

Thanks for taking the time to review my essay. I incorporated your suggestions as best as I could, tidied up citations and my use of Haida language.  I updated the discussion on theory, but didn't get too detailed as I want the essay to focus on a Haida approach. 

In appreciation,

LB